# Conveying Intangible Cultural Heritage in Museums with Interactive Storytelling and Projection Mapping: The Case of the Mastic Villages

Vasiliki Nikolakopoulou [1,*] , Petros Printezis [1], Vassilis Maniatis [1], Dimitris Kontizas [2], Spyros Vosinakis [1] , Pavlos Chatzigrigoriou [1] and Panayiotis Koutsabasis [1]

[1] Department of Product & Systems Design Engineering, University of the Aegean, 841 00 Syros, Greece; dpsd16095@syros.aegean.gr (P.P.); dpsd16060@syros.aegean.gr (V.M.); spyrosv@aegean.gr (S.V.); pavlos.chatzi@aegean.gr (P.C.); kgp@aegean.gr (P.K.)
[2] Kontizas Sound/Stage Mixing Engineer Freelancer, 841 00 Syros, Greece; jameskody.ost@gmail.com
[*] Correspondence: v.nikolakopoulou@aegean.gr

**Abstract:** Spatial Augmented Reality (SAR), as implemented with projection mapping, is part of mixed-reality technology with numerous applications in the cultural domain. In museums, interactive projection mapping has been exploited to superimpose virtual content on exhibited artefacts, offering users various hybrid ways to interact with the artefacts' physical and digital content. For this reason, it has been widely used in the context of architectural heritage to promote culture and raise awareness about historical buildings or landscapes by visualizing significant elements they convey. This paper presents the design, development, and iterative user evaluation of an interactive projection mapping installation for the Mastic Museum on Chios island in Greece that promotes UNESCO's intangible cultural heritage. The installation affords tangible interaction to activate the video projections presented in a storytelling manner on a 3D-printed scale model of a representative historic settlement exhibited inside the museum. The concept of this installation aims to connect the tangible and intangible cultural heritage of mastic and the related villages with narration and vivid illustrations. Three evaluation phases took place during the development at the lab and the museum, informing UX, learning, and design considerations.

**Keywords:** spatial augmented reality; projection mapping; interactive storytelling; museum; user experience; intangible cultural heritage; architectural heritage; mastic; UNESCO

## 1. Introduction

The concept of cultural heritage was revolutionized several times during the last decades of the 20th century [1]. The need to define and protect heritage in a more diverse worldwide context beyond national dimensions resulted in the adoption in 2003 of the Convention on the Safeguarding of the Intangible Cultural Heritage (ICH) by UNESCO, featuring its main elements [2].

Unlike tangible objects of culture, such as monuments, statues, artefacts, paintings, or books, ICH encompasses all the immaterial manifestations of the inherited practices that people perform as part of their everyday lives [3]. These manifestations represent humanity's living heritage, highlighting cultural diversity. The ICH of a particular community or group of people is characterized by the practices' identification [4]. This constitutes a prerequisite that ascribes cultural identity to the founders and carriers of these inherited habits. Moreover, as Bortolotto argues, "the main innovation suggested by the new definition of heritage proposed by UNESCO as 'intangible cultural heritage' does not rest on the intangibility of cultural expressions, but rather on its support of the idea that they are to be understood in terms of time (as an evolving process) and usage (not just for aesthetic contemplation)" [5] (p. 21). Therefore, the continual recreation of these cultural

expressions should be related to the historical and social evolution of the communities and groups concerned, as well as their cultural identity and authenticity. These relations value, acknowledge, and meet with human rights [4].

The new adoption of an advanced concept to define and preserve cultural heritage now considers more anthropocentric values than the initial regularization of historical and artistic ones [1]. Moreover, since heritage is no longer defined solely by its material nature, cultural value, the value of identity, as well as the ability of an object to evoke memory arousing the respective values have been added in the attempt to adopt a less normative approach in order to holistically document, safeguard, and appreciate the world's cultural heritage [1,6,7]. Furthermore, some researchers have questioned the sustainability of tangible cultural heritage if intangible heritage is not deliberately and similarly protected [8,9]. For example, historic areas or places are characterized by intangible values that can be physically and visually experienced and spiritually perceived [8,10]. In return, ICH is related to a place in terms of (1) the pattern of the area in which it appears, (2) the inhabitants' life, and (3) the craftsmanship, handcrafts, folklore, etc., that the inhabitants perform [8] inside that area. All these aspects contain human activity to be realized; a place's rural or urban landscape pattern results from human presence and activity [10,11]. Hence, there is a continual entanglement between human practices and their place of action, which are determined by each other. To understand each, one could reflect on how they are mutually presented in terms of their traces on the visible built space [12]. Consequently, and as Yuan (2008) poses, "any tangible culture must be supported by intangible value, and any intangible culture must rely on the tangible to be visualized" [7] (p. 8).

As the progressive concept of ICH endows the cultural heritage discourse, museums can also benefit from this approach. The holistic and people-oriented understanding of conceptualizing the past can establish enduring relations with the participation of communities as the leading carriers of ICH, thus making cultural representations reflect not only objects of culture but the people of these communities per se and their lives [13]. The recording of traditions with the contribution of a broader social web associated with the transmission of knowledge (customs, rituals, artistic expressions, craftsmanship) has already become the subject of modern museological research [14–18]. In addition, digital technology lessens the efforts of promoting ICH and facilitates the recording and survival of customs and traditions to ensure their sustained development in times of intensifying globalization [19].

Besides this, the digital transformation of museums into hybrid–complex and sensory enriched spaces is already a reality [20–22]. A multimedia approach, resulting in dynamic visualizations on surfaces and objects [23–25], can alter the visitor's perception of heritage based on an artistic, abstract, occasionally interactive, and more affective way of communicating the intangible values [26] outlined above. The dynamic representations can transcend the visitor's cultural experience by augmenting the exploration of heritage artefacts and historical events [27] or by connecting symbolic illustrations through vibrant narratives [28], thus enhancing cultural learning and supporting social interplay among visitors. In addition to the latter, the digitally enhanced exhibits that afford tactile engagement contrary to the *do-not-touch* outdated museum standard [29,30] also encourage public interaction, thus placing the museum closer to its audience and fulfilling its foundational social scope.

The cultural domain has applied Mixed Reality (MR) technology [31] in various contexts to immerse audiences in a combination of real and digital heritage content, providing users with numerous hybrid ways to interact with it [23,32]. Spatial Augmented Reality (SAR) [33], also known as Video Mapping or Projection Mapping, is part of the MR continuum [31]. SAR is mainly characterized by its ability to transform any surface or one or more objects into a screen through the projection of videos or images. It is also widely used in the field of architectural heritage [25,34–37], where storytelling content or artistic representations are projected on objects or buildings. The advantage of using this type of technology in the specific field is related to both the spatiotemporal information it allows to

be augmented on buildings (or their scale models) and the fact that it favors the concurrent participation of multiple users in a shared and sensory-enriched visual space. Nevertheless, SAR comes also with some limitations and challenges regarding the complexity of the interface the user interacts with [38–40]. In a museum context, the benefits of projection mapping could allow it to act as a heritage mediator [41] between visitors and cultural content, where projected videos on an architectural scale model would convey messages about tangible values and ICH. In this way, the values that theoretically integrate into the built visible space [12]—in this case, an architectural scale model of a historic settlement enriched with UNESCO enlisted ICH—could be visualized.

Although a considerable number of projection mapping installations have been designed and implemented inside museums to promote architectural heritage and showcase related spatiotemporal transformations [42–45], they do not extensively refer to their lab studies and how the final design has been reached. In addition, in a museum context, where there is no "experimental space" to iteratively evaluate the installations' performance during the various intermediate technological stages and design phases, e.g., from early prototypes to the refined set-up, the communicated cultural heritage content and the technology used could only be hypothetically aligned with the final visitor experience. To investigate these issues, we follow an iterative design approach exploiting various evaluation methods and tools to communicate ICH, as presented inside the museum, in connection with the tangible (architectural) heritage it relates through interactive projection mapping.

Therefore, in this paper, we present the design, development, and iterative user evaluation of a projection mapping installation for the Mastic Museum (https://www.piop.gr/en/diktuo-mouseiwn/Mouseio-Mastixas/to-mouseio.aspx, accessed on 14 April 2022) on Chios island in Greece, which displays historical events and seasonal cultivation activities on a 3D-printed scale model of a representative historic settlement of Southern Chios exhibited inside the museum. The installation affords tangible interaction to activate the video projections that are presented in a storytelling manner. A double set of 3D-printed, tangible artefacts, placed on the opposite sides of the scale model, activate either the historical events that took place on the island or the cultivation activities the people perform throughout the year as the main part of their ICH. The concept of this installation aims to connect tangible values and ICH with narration and vivid illustrations on a scale model that represents the visible part of the ICH of the place concerned. Considering that the installation is positioned at the end of the museum visit, we assume that visitors would have a concluding opportunity to recapitulate and reflect on the intangible values that the museum is aiming for the visitor to perceive. The design approach and development of the interactive system presented in this paper have been composed and processed in the context of this hypothesis.

The paper is structured as follows. In the first section (Introduction) after a brief literature review in the relevant field, we define the purpose, the objective and the basis hypothesis of the present article. The following section discusses the related work in projection mapping installations and interactive exhibits focused on (intangible) cultural heritage and architectural heritage, which are either inside a museum or at a historic place or building of related spatiotemporal reference. In Section 3, we introduce the Chios Mastic Museum, the UNESCO's ICH of 'mastiha' (mastic) cultivation, and the exhibits that are associated with our projection mapping installation. Section 4 presents the detailed concept and the design and development of the installation, whereas Section 5 describes the user evaluation process and results. Section 6 discusses the outcomes and limitations of this study, reflecting on lessons learned, and declares its conclusions.

## 2. Related Work

Cultural heritage museums and sites, as well as museums in general, have been considerably favored by the use of MR technology and SAR specifically [23–25]. The ability to superimpose virtual content on real exhibited artefacts on the floor and to let users interact with it through various interfaces has revealed a new style of engaging

museum audiences in the deeper sense-making of the exhibited content. Research so far indicates that SAR has been adequately used in cases where (simple) physical interaction is demanded since users do not need to wear any special equipment. In addition, in cases of collaboration and mutual communication among multiple users when in a shared projected environment, SAR has been similarly utilized. For instance, in the work of [38], the case of a floor-based User Interface (UI) is explored to immerse and guide pairs of users in a shared environment that projects scenes of a dinosaur skeleton. In a similar work for a museum's zebrafish model [39], the use of kinesthetic interaction through hand movements and shadows to display information on the model is investigated to understand the users' enjoyment and the intuitiveness of the selected UI. Interestingly, as in the work of [46], which fuses projection mapping with camera-based interaction techniques, it seems that SAR holds the element of surprise, while, at the same time, its ability to place the interactive experience clearly in the physical world enhances this element and stimulates users' engagement and enjoyment with the projected content. Nevertheless, researchers express their concerns about the ease of use and learning of such applications when more information or guidance is added in the respective interface or when obstructive users' shadows appear. Finally, and from a more technological perspective, experimentation with the use of multiple projectors to display information in order to overcome shadows or limited angle views [31] (p. 88) has even allowed user's personalized viewing angle of the projected material [47,48].

Most of the projection mapping installations that appear in cultural heritage museums involve the digital augmentation and exploration of exhibits. In [40], researchers use an effective and simple interaction paradigm of a flashlight to reveal information about eroded stones. So, users can point to particular areas of the heritage artefacts for a more detailed inspection that is virtually supplemented, thus maintaining a link between the real object and its virtual expansion. Similarly, in the work of [49], animation, text, and visual effects are projected on a Danish rune stone, which constitutes a permanent installation at the respective museum. In this installation, the researchers experiment with various ways of blending layers of projection with the respective interactions that the visitors perform in order to attract their attention and finally engage them in playful and meaningful conversations. Another projection mapping installation that uses off-the-shelf components and affordable technologies to tell the story of a historic electric car has been tested at the Syros Industrial Museum in Greece [50]. There, an interactive wall activates the projection of related content through touchpoints of conductive paint that guide the visitor from one animated narration to another, unfolding the whole story of the car.

Other kinds of museum exhibits that encourage projection mapping techniques are scale models of historic cities, landscapes, and buildings. As presented later, there is an essential connection in the field of architectural heritage with the use of video mapping, which has also lead to the coinage of the term "architectural mapping" [36].

Architectural mapping may take place indoors or outdoors. In the museum context, we come across several installations that interactively project information onto scale models based on the authentic models' historical and architectural documentation. For example, in [42], an interactive installation of a 3D-printed model of the World's Heritage Site Fray Bentos industrial landscape is used inside the museum of Industrial Revolution in Uruguay, allowing visitors to explore projected architectural information in the form of a tour. Similarly, in [43], an enormous 3D-printed model of Fort Saint-Jean installed at the Musée des Civilizations de L'Europe et de la Méditerranée (MuCEM) in Marseille allows visitors to take a virtual journey by combining various projection methods dispersed in several rooms of the museum. The same immense effort on 3D printing, video mapping, and heritage information organization and management through a touch screen interface is installed at the Nantes historical museum to provide visitors and historians with detailed projections about the local harbor's historical activity [44]. All the three works report on the multidisciplinarity of the documentation and development teams involved and reflect on the challenges of the technologies used, the digitization process, the storytelling, and

the 3D printing of the models. In addition, all researchers argue in favor of the educational and learning potential that such efforts integrate for audiences while at the same time representing an effective means to disseminate and promote cultural heritage.

Likewise, architectural mapping that occurs outdoors is correspondingly used to raise further awareness of the vulnerability and significance of the cases studied. The experimentation of video mapping on mock-ups for the Ascoli Piceno architectural heritage in Italy is presented by [45] for restorative, narrative, communicative, artistic, and educational [51] purposes. With a less performative focus, in the work of [35], researchers make use of a Tangible User Interface (TUI) to spatially drive visitors onto the respective projections that unravel the spatiotemporal transformation of a medieval chapel that have occurred during the last 850 years. The installation was evaluated in situ, and findings report on the facilitation of visitors' acknowledgment and memorability of the chapel's aesthetic features, while the TUI seems to have enhanced both the exploration of these features and communication among the visitors. Another work, which combines SAR and mobile AR technology to raise awareness of the artistic, historic, and cultural uniqueness of Basilica of Saint Catherina of Alexandria in Galatina [36], aims to provide the visitor with a distinctive digital journey while also integrating the visual qualities of the place. In this work, researchers cautiously and progressively adopt the affordances of the two technologies to promote and enhance the understanding of the buildings' characteristics. Again, the educational potential is indicated in relation to the attractiveness and stimulation UX evaluation obtains. Finally, in [37] according to the concept of 'genius loci' (https://www.oxfordreference.com/view/10.1093/oi/authority.20110803095847893, accessed on 14 April 2022), the intangible quality of a place [10] is approached through a multidisciplinary project that is mainly based on precise 3D data acquisition of the charterhouse of Villeneuve-lès-Avignon (Gard, France), the related tools and documents, and their final 3D hypothetical restitution with the use of SAR in situ. The researchers present the implementation workflow of the individual digital features, such as anamorhic projections, visual effect renderings, spatial and audio staging, and how they might contribute to the visitors' immersion in the "spirit of the place".

Consequently, projection mapping in the case of architectural heritage seems to be interlinked with the communication of intangible values that a place acquires and could be rendered via its architectural features: their evolution in time and usage. Georgescu Paquin [41] argues for the transcending mediation that projection mapping can enable for building in terms of adding a message to it—be it artistic, exploratory, explanatory, or annotated through audiovisual enrichment.

On the other hand, it would be interesting to review a few relevant applications that explore and promote ICH content in their interactive applications, which are not necessarily projection-based. Meaningful work is found in [26], where researchers follow value-sensitive design to develop a museum tabletop installation dedicated to Musqueam culture. Users place symbolic contemporary and ancient belongings from the living culture onto the tabletop, and activation rings display related information supplemented by another three monitors. The installation is designed first to evoke the visitors' attention and then engage them in inspecting the belongings, unlocking special stories about the culture's intangible heritage. The central values communicated are recognition and respect for the preserved knowledge that the culture continues to deliver. In the same context, but less symbolically, the work in [28] presents two case studies based on interactive tabletop narration using tangibles. The aim is to contextualize the meaning of the representative cultural tangibles and help visitors to connect personally with them. The researchers propose a narration framework for other designers and museum practitioners to follow, as in [52], and focus on mapping the interactions' complexity with the aims that the multimedia content tries to achieve. Complementing the aforementioned multidisciplinary works consisting of various professionals, in [53], we find the co-designing activities and iterative prototype evaluation of a playful MR museum installation of an interactive crane that aims to promote the ICH of Tinian marble crafts. The installation places the visitor in

the role of a crane operator inside a virtual historic quarry with the goal of communicating the intangible aspects of the Tinian craftspeople's working culture. Finally, a promising look-alike CAVE laboratory setup that exploits projection mapping, physical computing, and storytelling for historical beekeeping at the Cyclades in Greece is presented in [54]. The designers developed and evaluated four distinct usage scenarios to engage users in an edutaining and playful storytelling approach that was later exhibited in public.

In the presented work, we design, develop, and iteratively evaluate the UX and subsequently the learning effect of an interactive projection mapping installation for the Chios Mastic Museum in Greece that uses storytelling to unfold the island's history, which is decisively related to the ICH of 'mastiha' and its precious cultivation knowledge. In relation to the research presented so far, we utilize an architectural scale model of a representative historic settlement exhibited inside the museum to act as a canvas for stories projected in 3D animation videos with audio narration. The historic settlements of Southern Chios, the mastic villages or *mastihochoria*, constitute determinant places that acquire intangible values of the inherited culture of the mastic cultivators who have inhabited the settlements for hundreds of years. Apart from the intangible values, the settlements' architectural evolution also manifests the history of the turbulent years and the alternation of sovereign regimes on the island throughout the Middle Ages. Subsequently, the mastic heritage has deeply penetrated the social, economic, and political development of the island and the mastic villages. Consequently, a holistic approach to illustrate this impact should include stories and references to various aspects, including historical events, architecture, and cultivation tools and processes.

The approach proposed in this paper exploits interactive storytelling, SAR, physical computing, and 3D printing to present an inclusive view of the impact that ICH can have on a place and vice versa. In this case, the related spatiotemporal transformation of a historic settlement encompasses the intangible values of the inherited practices that people perform in this place. The stories illustrate with audiovisuals the crucial historical periods, facts, seasons, and tools, which are triggered by visitors through 3D-printed tangible artefacts to project the stories onto a scale model that has also been printed in 3D. Therefore, the following paragraph introduces the museum and its exhibits that were crucial to form the concept model of our interactive installation.

### 3. The Chios Mastic Museum: Exhibiting the Intangible Cultural Heritage of Mastiha

The Mastic Museum, harmonized with the natural landscape of the island of Chios, is located near the mastic village of Pyrgi. It is dedicated to the intangible cultural heritage of 'mastiha', inscribed in the Representative List of Intangible Cultural Heritage of Humanity by UNESCO in 2014 (https://ich.unesco.org/en/RL/know-how-of-cultivating-mastic-on-the-island-of-chios-00993, accessed on 14 April 2022). The heritage of Chios mastic, a unique resin "weeping" out of a shrub with numerous medicinal properties, encompasses a traditional culture where members of all ages of the entire family are occupied with its laborious cultivation and production throughout the year. The knowledge of cultivating the mastic requires the knowledge of the techniques for incising and harvesting the tree and is transmitted from one generation to the next. Over the ages, this culture has founded a network of community associations for the cooperative exploitation of mastic based on mutual support, thus representing a social affair. The collectivist memory of the culture of mastic has penetrated over the years into the communal practices of the people of Southern Chios, who are the main bearers of this unique heritage.

In 2016, the Mastic Museum was the 17th most visited museum in Greece, according to the IOBE data (http://iobe.gr/docs/pub/PRE_20052017_PUB_GR.pdf?_x_tr_sch=http&_x_tr_sl=auto&_x_tr_tl=el&_x_tr_hl=el&_x_tr_pto=wapp, accessed on 14 April 2022), competing with other famous civic Greek museums, such as the museums of Acropolis, Delphi, and Ancient Olympia. The museum highlights the sustainability of the mastic tree and its traditional cultivation, processing, and industrialization. At the same time, it aims to integrate mastic's significance into the broader social, economic, historical, architectural,

and cultural scenery of Chios. By including traditional mastic cultivation in the representative UNESCO list, emphasis is put on the timelessness and the diachronous quality of the product. To summarize, the goals of this museum are as follows:

1.  The manifestation of the traditional cultivating knowledge through exhibits (traditional costumes, tools and equipment), archives, images, and videos of activities of the people of mastic villages.
2.  The interpretation of the historical management of such a unique product and how it has affected the rural and inhabited landscape of the southern region of Chios and its settlements.
3.  The promotion of mastic's cooperative exploitation and administration in modern times and the various alternations that Chios mastic has adopted before its global exportation with the support of a creative network initiative of mastic product fabrication.

The permanent exhibition deals with mastic as a decisive local natural product, and it is organized into five main thematic sections, as seen in Figure 1.

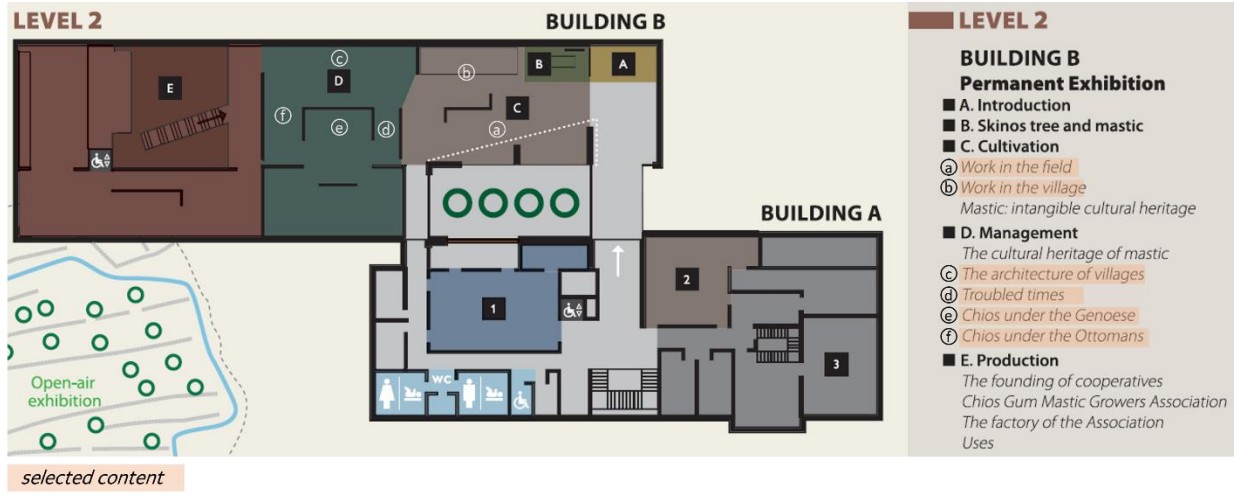

**Figure 1.** The Chios Mastic Museum map and appearance of the selected content in orange. © Piraeus Bank Group Cultural Foundation.

### 3.1. Mastic Cultivation

At the beginning, in sections A. and B. (Figure 2d,c), the visitors can learn about the special variation of 'pistacia lentiscus var. Chia', grown only in a specific region of the world. Then, the principal exhibition, section C. (Figure 2a,b), is dedicated to the traditional knowledge of mastic cultivation, with exhibits about the works in the field, the tasks in the village, accompanying traditional songs, and the main tools and equipment used, represented by large images, videos, and installations. The section describes mastic's cultivation and production phases, showcasing the transmission of skill from generation to generation and how the local society is organized around mastic's life cycle.

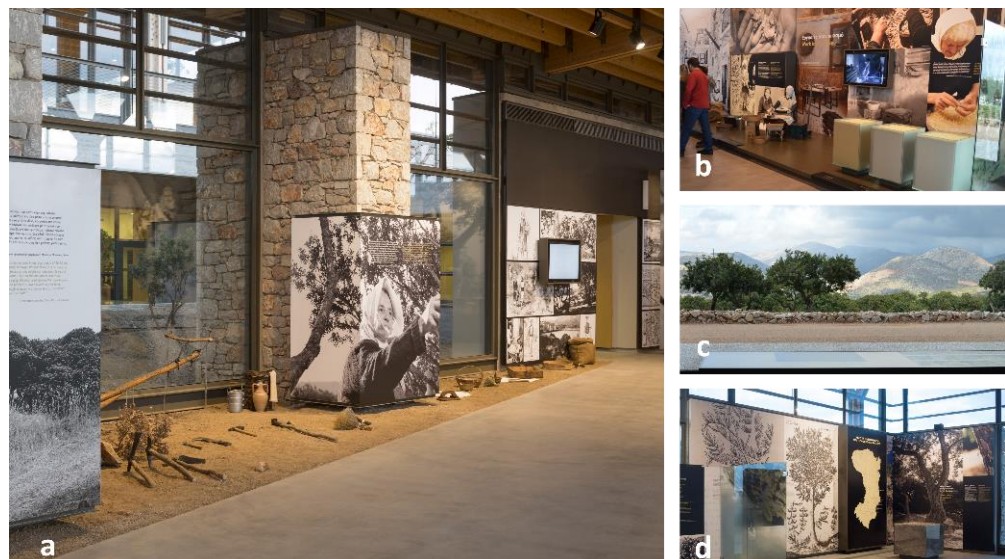

**Figure 2.** The mastic cultivation exhibition at the Chios Mastic Museum (Sections A, B, C): introductory folklore song (**d**) and botanical information (**c**), and cultivation activities in the field (**a**) and at the village (**b**).

The cultivators should take care of the tree throughout the four seasons. The tree's treatment begins in the winter, then cultivators clear and prepare the ground, while in the late spring and the beginning of the summer, they perform cautious incisions at the trunk and the branches in order for the tree to "weep" the resin. After some days, the resin has flown, and then the gathering of the *mastiha* starts. When the weather becomes cold, the women cultivators process the mastic in the village. They sieve, wash, and clean the mastic "tears" to provide them to the Chios Gum Mastic Growers Association: the modern cooperative exploitation of mastic that takes care of the final production phases.

### 3.2. Management of Mastiha

In the following exhibition, section D. (Figure 3), the historical management of 'mastiha' is showcased alongside the way it has shaped the rural and inhabited landscape of the island's southern region and its cultural heritage since Chios was part of the Byzantine Empire up to its final integration in the Greek state. During these ages, the intermediate Genoese and Ottoman overlordship regimes systematically exploited the resin. This systematization formed the southern region's rural landscape by rearranging the settlements, building new houses, and forcing the social life of the mastic cultivators to adapt to the changes. The Genoese, intending to regulate the mastic cultivation, fortified the settlements, as for them, the mastic cultivators constituted a special category of reliant peasants settled inside the village fort. Under the Ottomans, mastic cultivators were given some privileges, revealed by the dense tissue that the settlements present during these times, and the development and advancement of the housing. It then becomes apparent that the uniqueness of 'mastiha' led it to become a product of systematic cultivation and exploitation by the respective rulers of the region throughout the MA and modernity. It constituted a determining factor for the economy of the island.

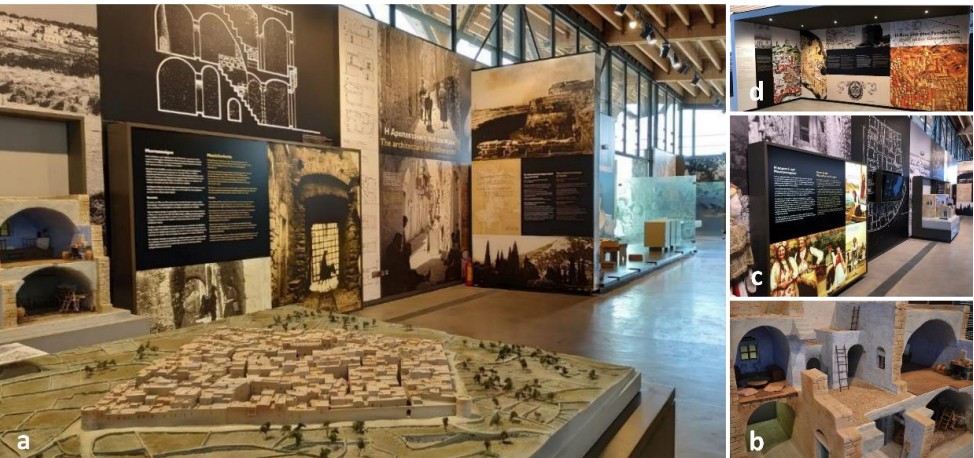

**Figure 3.** The mastic management exhibition at the Chios Mastic Museum (Section D). A scale model of the representative mastic village Olympoi (**a**) at the center, a model house (**b**), architectural documentation in large drawings and audiovisual (**c**), and textual information about the various sovereign regimes (**d**) at the back of the scale model.

In this museum section, there are various forms of exhibits that narrate mastic's historical management: two scale models, one of a representative settlement with a rural area (Figure 3a) and one of the interiors of a house (Figure 3b); a 3D-animated video presenting the architectural evolution of the mastic villages and architectural drawings (Figure 3c); many large images from the villagers' everyday life inside and outside the settlements; traditional costumes; historical artefacts; much textual information about the historical events related to the alternation of the sovereign regimes (Figure 3d); and a 20 min inclusive and recapitulative video summarizing the history of 'mastiha' from the 1st to the 21st century.

### 3.3. Merging the Two Exhibitions

The exhibition about mastic management aims to expose the visitor to the difficult living conditions of the people of the mastic villages, mainly by using historical and architectural documentation. Starting with the Genoese formation of the settlements as a fortification structure, it becomes apparent that the central castle, the exclusive gate, the wall, and the narrow streets among the dense construction of the houses constitute a way of controlling the mastic villagers and thus the production of the mastic. At the same time, and in alliance with a fortified coastline system along the island that could communicate with the central castles in each settlement, it was a way to be informed and protected from would-be intruders of that time—mainly pirates. The visitor can recognize the agricultural (and livestock) activity of the inhabitants from the premises of the model house (Figure 3b), such as the ground floor, which serves as a stable and a warehouse. This marks a coverage of their basic living needs while also indicating the living settings and substandard hygienic conditions. For instance, there was a coexistence with animals and a lack of ventilation and lighting due to the structural density of the houses, especially during the Genoese period where only ground floor buildings existed, and these factors were responsible for diseases (mainly tuberculosis) and the transmission of epidemics (such as the plague).

In this heritage museum's dimension, bindings between culture and environment exist alongside the landscape and the people's natural heritage. In this entanglement, over time, people and nature tacitly adopted and tailored a cultural heritage committed to 'mastiha' throughout the succession of the sovereign regimes, the turbulent times, and the difficult living conditions caused by the suffocating control and discipline of the cultivators' productive liabilities.

## 4. Design and Development

The aim of the permanent exhibition, which melds the natural landscape of the mastic villages with the mastic's cultivating activities, life, and customs of the people, is to mold the concept of the interactive installation that intends to communicate the intangible aspects of the heritage involved. Architecture, as an element of culture defined by landscape and human activities, in our case, stands as the pivotal axis affected by both the mastic people's cultivating habits (buildings and house floors dedicated for these habits) and the structural reinforcements made by the various regimes (fortified urban evolution of the mastic villages). The scale model of the mastic village found in section D. (Figure 3a), the Olympoi settlement, can then act as a reference point to project a two-faceted story:

1.  One facet depicts the troubled years, the Genoese and Ottoman overlordship regimes, and the historic cooperative mastic exploitation that advanced the industrialization of the island and Greece in general; and

2.  One facet shows the repeated seasonal occupations of the mastic villagers to cultivate 'mastiha', and their main activities and representative tools for each season: from treatment to the final cleaning of the mastic "tears".

Therefore, our concept is to provide visitors with an interactive scale model onto which they can trigger projections to unrestrictedly unravel all the relevant information of the surrounding exhibits that encompass the two-faceted story. The information should unfold in a storytelling manner, triggered by symbolic artefacts related to the story's individual elements.

### 4.1. Concept Overview

The installation's function is to display information in the form of short stories on the scale model. The information is multi-layered and essentially concerns two categories. On the one hand, the historical information projected through the scale's architectural elements refers to the social, cultural, and historical conditions and how they affected the settlement and, consequently, the inhabitants' lives. On the other hand, the mastic cultivation cycle, which has been constant over the years, is the central element of the inhabitants' life. Therefore, the storytelling axis is shared in two independent "speed-ways", as illustrated in Figure 4. The linear one belongs to the historical periods related to the island's conquerors that influenced the settlement's architecture. The cyclical belongs to the seasons' alternation during the year, which affects the mastic's cultivation cycle and its processing and storage.

Visitors can interact with the installation through tangible objects, which they operate in pairs on two stations placed at the opposite sides of the scale model: one referring to architecture and one to cultural activities. The tangibles then indicate the corresponding physical objects, such as architectural elements, conquerors' symbols, and representative seasonal cultivating tools. After that, their placement in a specific station area alters the flow of information by providing the respective audio-visual story (https://www.youtube.com/watch?v=gTV9GXSv2Ws, accessed on 14 April 2022).

Accordingly, the two-faceted storytelling concept organizes elements resulting from research on the visiting experience, the surrounding exhibits, their indexing, the projection mapping affordances, and the connections that appear in the context of the hypothetical merging of tangible (architectural) and intangible aspects that the museum's curation aim to provide. The elements of the museum, the concept model, and the installation are organized in the form of requirements in Table 1, which helped the design team to create their visualization and the development of the overall interface.

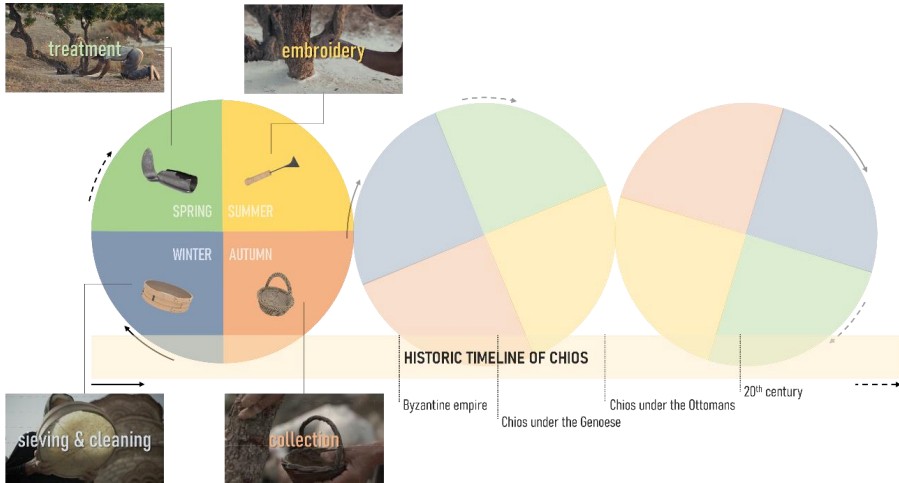

**Figure 4.** The storytelling concept of the two-faceted story: historical (changing architecture) and seasonal (cultivation).

**Table 1.** The museum's and concept model's elements and how they appear in the installation.

| Elements | | |
|---|---|---|
| **Museum** | **Concept Model** | **Installation** |
| Regular museum visits: at least in pairs | Users (at least two) | Two (interactive) stations |
| Mediating tangible values and ICH of the mastic's people | Settlement's 3D scale model | Shared in two parts: architectural, rural |
| Architectural evolution manifesting mastic villages' history | 3D model: architectural features | Central castle, exclusive gate, wall, narrow streets |
| Landscape as the natural heritage of people affecting architecture | 3D model: rural features | Terrain, mastic trees, trails |
| Four critical historical periods related to island's sovereign regimes | Station: architecture | Four crucial historical events |
| Four main seasonal mastic cultivation activities | Station: cultivation | Four seasons of the year |
| Objects representing architectural evolution and found inside the museum's exhibits in the Management section, D. | Tangibles: history & architecture | Turret (Byzantine years), medieval soldier (Genoese), minaret (Ottomans), industrialized mastic gum (20th century) |
| Objects representing cultivation activities and found inside the museum's exhibits in the Cultivation section, C. | Tangibles: cultivation | Shovel (Spring), prick (Summer), basket (Autumn), sieve (Winter) |
| **History and architecture stories** | **Tangibles** | **Visuals** |
| Villages' central castle, its purpose and connection to fortified coastline system from early Byzantine years | 1. Turret | Pirate alert from fortified coastline system's turret, highlighting central castle, churches, pirates' failed invasion |
| The Genoese militarization approach and controlled cultivation led to villages' structural fortification | 2. Medieval soldier | Architectural evolution from central castle to the wall, first houses, narrow streets, gate |
| The Ottomans' providence to mastic cultivators led to villages' houses expansion, but later to vengeance (connection to the 1821 Greek revolution) | 3. Minaret | Spread of settlement's houses to the wall, houses' floor evolution, the massacre of Chios |
| The island's new industrialized era (20th century) is depicted in the Production section, E—villages' expansion, mastic's production and exportation, Chios integration to the Greek state | 4. industrialized mastic gum | Houses outside the settlement, wall demolition, industrialized mastic gum production and exportation, Greek flag's placement |

**Table 1.** *Cont.*

| Elements | | |
|---|---|---|
| **Mastic cultivation circle activities** | **Tangibles** | **Visuals** |
| Springtime treatment, clearing soil around the tree with the shovel, hands, and broom | 1. Shovel | Spring terrain, mastic tree treatment activities, other basic tools and cultivation actions |
| Summertime white soiling around the tree, then embroidery with pricks by performing incisions at the tree's trunk and branches | 2. Prick | Summer terrain, tree embroidery, other basic tools and cultivation actions |
| Autumntime collection of mastic tears with basket, "big tears" first, then cleaning with broom, sieving, and gathering with wooden chests | 3. Basket | Autumn terrain, mastic collection, other basic tools and cultivation actions |
| Wintertime repeated sieving, washing in cauldrons, drying, and final cleaning (biting) of mastic tears | 4. Sieve | Winter terrain, settlement areas also activated, mastic sieving and cleaning, other basic tools and cultivation actions |

In the following two sections, we present the installation's overview and use and the issues related to implementation and content production, construction, and development. We describe how the installation elements are processed in terms of graphic design, interface design, 3D printing, video mapping, sound design, and system architecture. Thus, we present their final look, while in Section 5, we refer to the alterations they underwent through an iterative user evaluation procedure to reach the desired outcome.

*4.2. Installation Overview*

Following the legends in Figure 5 (left), the system consists of a table (1) on which the 3D model (2) is placed. On the right and left of the table, the two stations (3) house a touch screen, four tangibles, and an NFC reader placed below an activation area. Under the table, we find all the computer equipment (4) together with the audio system (5). Above the 3D model and mounted on the top of the installation lies the projector (6).

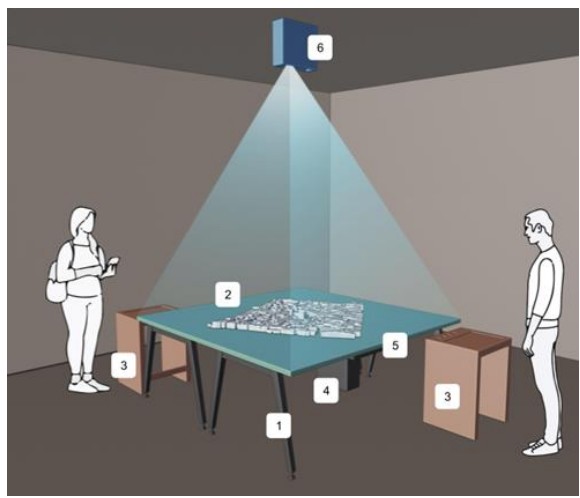 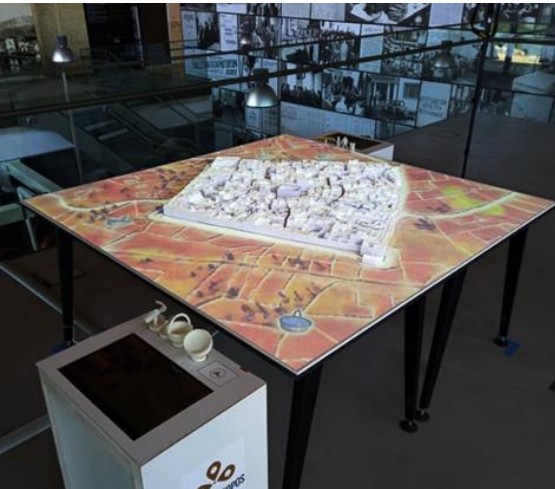

**Figure 5.** (**Left**) The installation's spatial system architecture: table (1); 3D model (2); interactive stations (3); computer equipment (4); audio system (5); projector (6). (**Right**) Final system's installation at the Chios Mastic Museum.

The installation runs on a desktop computer with an internal graphics card with three screen outputs: one for the projector and two for the stations' screens. Also connected to the computer are two USB cables for the touch screens, two NFC readers, and a stereo audio system.

When generally activated (Figure 5, right), the system enters a loop state and waits for a user to give input. Standing at a station, the user selects one of the four tangibles extended above the screen and places it on the activation area. By doing so, the NFC reader recognizes the tangible asset via the NFC sticker placed at its bottom. The station's screen and the projector display the corresponding projection upon identification. At this stage, and until the end of the projection, the user can only interact with the system via the "language" and "info" buttons (Figure 6, left). At the end of the projection, the system returns to the loop mode waiting for the following input. Each video projection lasts about 1 min, so if the users activate all projections, the time required to use the installation is around 8 min.

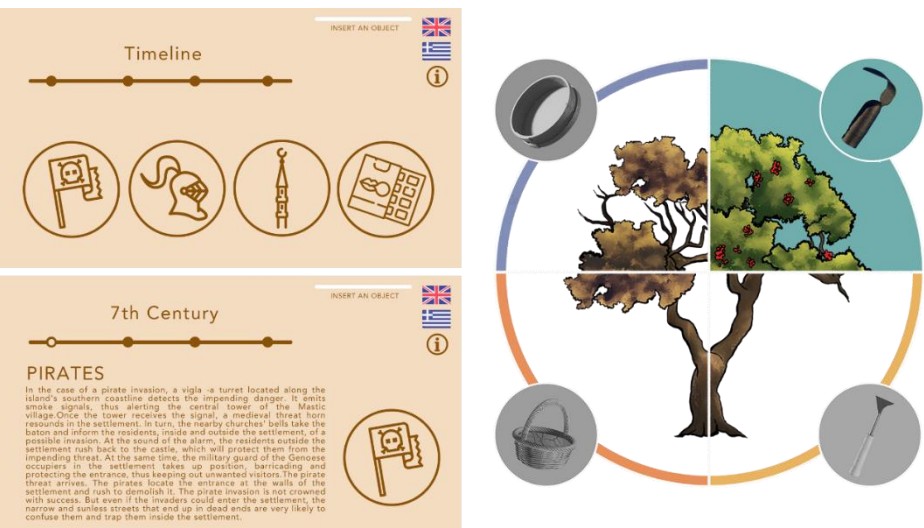

**Figure 6.** (**Upper left**) The historical events' station screen. (**Bottom left**) The text displayed for the activated pirate alert projection, listing the narrator's audio. (**Right**) The seasonal cultivation tree is divided into four, showcasing the selected spring projection with the relevant tool, e.g., a shovel.

The installation is mainly audiovisual. For the best possible understanding of the information, a narrator describes everything projected on the surface of the 3D model, thus turning the installation into an interactive documentary.

The station controlling the historical events contains the four tangibles referenced in Table 1. "History and architecture—Tangibles" and depicted in Figure 7. We show these four symbols extended and a timeline of events in Figure 6 (upper left). By activating a tangible asset, the timeline shows when the event occurred while a text is displayed below listing the narrator's audio (Figure 6, bottom left). Accordingly, on the opposite side's station controlling the seasonal cultivation activities, the four representative tools are depicted in Figure 8. On the screen, we find a 2D sketch of a tree representing the mastic tree, divided into four parts (Figure 6, right). Each quadrant showcases the tree in a different season. By selecting a tangible, the tree appears as in the corresponding period together with the relevant tool (Figure 6, right), while a text is displayed next to it listing the narrator's audio.

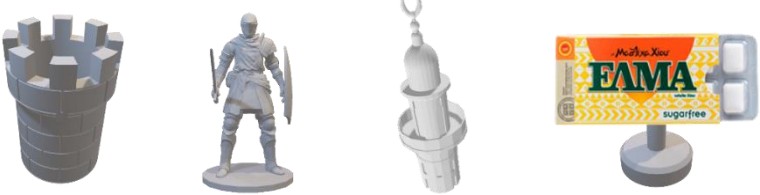

**Figure 7.** From left to right: turret; medieval soldier; minaret; industrialized gum ELMA.

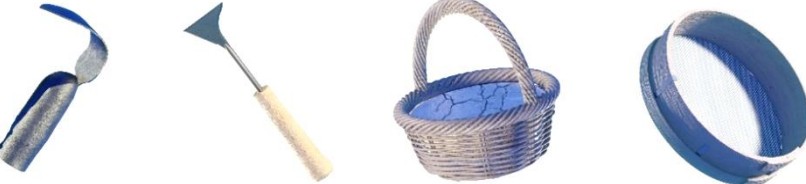

**Figure 8.** From left to right: shovel; pick; basket; and sieve.

In both stations, there is an indication for the activation area "insert an object", two buttons for selecting Greek or English language represented by the respective flags, and a button for instructions and more information (Figure 6, left) such as credits and acknowledgements. The narration was recorded with a male and female voice in Greek and English and integrated into the system. To give the finest and more appropriate character to the audio, our team went through recreating all the surrounding sounds with the Foley process [55], which is the recording of sounds and SFX and took place live in the original location with the use of original materials and objects. These sounds were isolated and adequately processed to cover the system's narrative and graphics. At the same time, this sound digitization process also contributed to preserving an essential acoustic part of the ICH involved. Finally, ambient sound, mumbles, voices, and music were added to complete the UX, transporting the user to a multi-sensory viewing environment with the intent of avoiding distraction from the basic information.

The projections are rendered in 3D graphics—mainly the historical events' projections—and in 2D graphics—mainly the mastic-cultivation's processes—for more effective communication of the narration illustrations.

### 4.3. Implementation

The installation revolves around the mastic village, and the whole development process starts with its composition. The settlement is three-dimensionally designed and emits the most realistic character possible (Figure 9, left). All other graphic elements were later designed and layered upon it in the form of 2D and 3D videos (Figure 9, right). This procedure defines the background of all videos to be created, delimits the viewing area, and helps to improve the accuracy of the following projection mapping activity. The main software used was Blender (https://www.blender.org/, accessed on 14 April 2022) and the Adobe (https://www.adobe.com/creativecloud.html, accessed on 14 April 2022) suite.

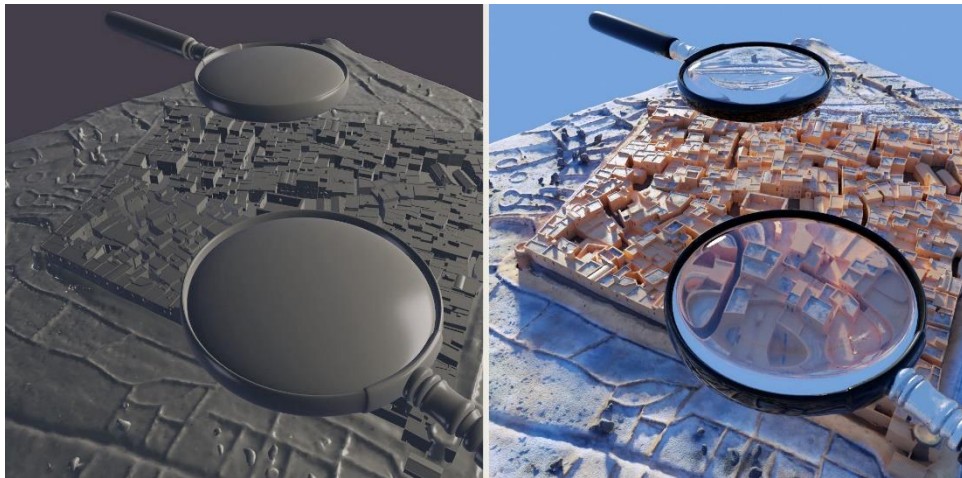

**Figure 9.** (**Left**) 3D model of the mastic village. (**Right**) Wintertime graphic elements layered upon the 3D model.

The 3D graphics allow historical events such as the pirate alert (Figure 10, left) and earthquakes and wall destructions (Figure 10, right) to be reproduced with historical, architectural, and topographical accuracy, thus focusing both on the visitor's excitement while maintaining their educational interest. For example, in Figure 10 (left), showcasing the pirate threat alarm, the smoke signals indeed come from this direction regarding the position of the settlement and the respective turret located alongside the fortified coastline system responsible for delivering the signals. However, this does not mean that design freedoms cannot be taken for the sake of either a better understanding of the story or the maintenance of the visitor engagement with the content. For instance, in Figure 10 (right), the Greek flag did not have this exact appearance during the historical period referenced in this projection.

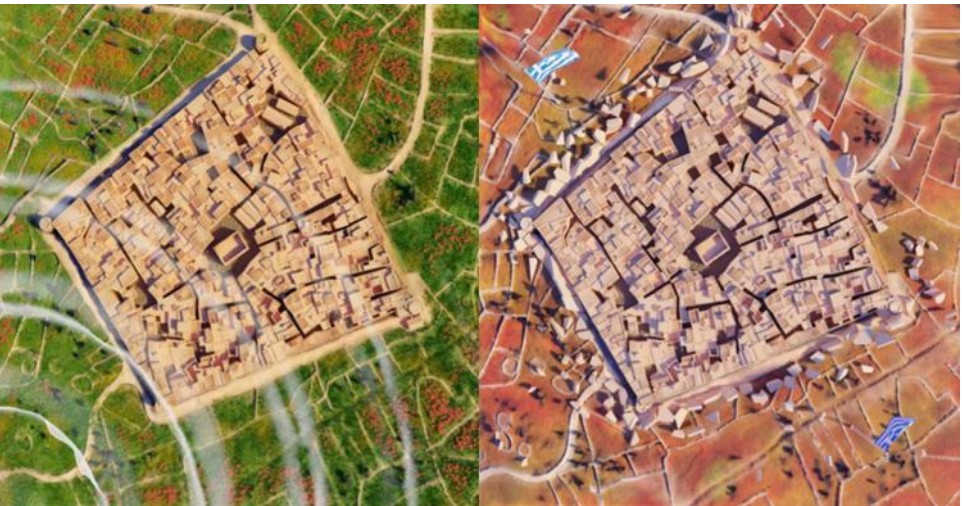

**Figure 10.** Pirate alert (**Left**), wall destruction (**Right**).

Speaking of realism, from a geographical point of view and regarding color and structure, the settlement (Figure 9, left) is an exact digital copy of the actual settlement provided by the museum administrators. By using this version, we contribute to preserving its natural and architectural heritage and integrating its values into our projected narration. Nevertheless, elements such as the terrain's seasonal look and the weather conditions have been creatively exaggerated. It is important for the visitor to realize the seasonal variation even from this unrealistic scenario, e.g., the winter terrain in Figure 9, right. Along the same line, dramatized events such as bloodshed in the Ottoman scenario (Figure 11, left), representing the historic massacre of Chios in 1822, and other symbolic elements, such as the blanket of the industrialized gum ELMA floating over the settlement (Figure 11, right), offer a spectacular view and keep the visitors engaged with the content.

On the other hand, the mastic cultivation projections involved more complex and tedious tasks concerning their design. The realistic depiction of the cultivation constituted a challenge that required laborious re-design. Having the cultivation activities already vividly presented in multiple parts of the museum's exhibits, we decided to simplify our illustrations in 2D graphical animations depicted via magnifying glasses (Figure 9). The human factor, such as the cultivators' hands included in the activities, was removed, and the product itself, the mastic and its tree, was left to narrate its course.

The stations' graphic design was created bearing in mind the museum's aesthetics. It is similar to much of the exhibits' legends, and their color palette was inspired by the installation's surrounding area. As in a legend, the information is kept flat and clear (Figure 6).

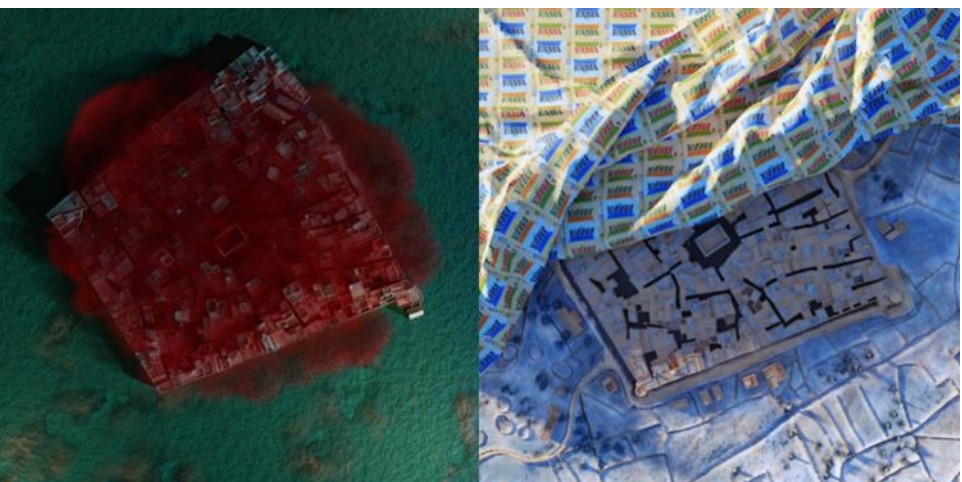

**Figure 11.** Bloodshed scenario from Ottoman rule (**Left**), ELMA gum blanket (**Right**).

Concerning the projection mapping and in the context of the original settlement's digital 3D model, we entered it into a digital environment and performed the related video projections there before projecting them onto the installation's final 3D-printed model. We used the Capture software to introduce the digital model and thus simulate the projection process. In this way, we identified the optimal viewing angle, the distortion required by the projector, and the distance the projector should have from the physical 3D model so that all these could be integrated into the finalized graphics without the need for re-mapping, re-calibrating, and testing each time during the trials. Furthermore, the effective communication between the graphic designer and the projection mapper helped the final projections to fit perfectly onto the 3D-printed model.

To finalize the installation, we first had to go through a prototyping process, elaborating the graphics and the overall installation assembly. Using programs such as Mad mapper (https://madmapper.com/, accessed on 14 April 2022) for video-mapping, connected to Qlab (https://qlab.app/, accessed on 14 April 2022) as a media player via a Mac computer, we created a complete, non-functional prototype that helped us to realize our system's capabilities. We thus managed to develop its final appearance and operation without initially entering into a coding problem, and this allowed us to make quick changes and re-designs.

Since the installation does not require projection on more than one surface, only one projector with specific characteristics was needed. For the spatial needs of the installation, the projector we finally used had a wide lens spectrum, high brightness for an already lit space, and afforded the possibility of vertical placement. The final system uses only one coding Unity file, which triggers the already mapped videos each time and loops the projector displays onto the 3D printed model. Whenever the PC is switched on, it runs the file without additional actions from the museum's staff.

The model's blocks of the houses were 3D printed in groups and placed on a square PVC surface of 160 cm. The whole 3D-printed model and surface are white so that we can project any desired texture and color onto them. The entire construction (model and surface) is placed on a square table of the same dimensions and a height of 80 cm in order for the users to have the finest possible view. At the same time, the space under the table is used to store the installation's equipment, such as the PC, cables, and speakers. The settlement is placed at a specific angle on the square PVC so as to create larger triangular spaces on the surrounding surface upon which more information and graphics about cultivation activities could fit.

In order to reach the desired outcome, we progressed through three distinct user evaluation phases, both at the lab and the museum, presented in the following section.

## 5. Evaluation Process and Results

We followed an iterative design approach to determine how users perceive our concept and how they respond to the various features integrated into the installation: tangibles, 2D and 3D illustrations, audio narration, displayed textual information, and interaction with the interface. An inclusive table of the methodology followed is provided below (Table 2). We first employed a formative evaluation at the museum, testing a low-fidelity (lo-fi) prototype with our stakeholders, i.e., the museum's curators, staff, members of the design and development team, and the visitors. After finalizing the prototype, we examined it in the laboratory with a small sample of users to verify its performance. Then, we installed the final version of the system at the museum, carrying out the final field study. All users consented to be video-recorded during the evaluation and to provide their demographic data.

**Table 2.** Iterative evaluation process for the Chios Mastic Museum interactive installation.

| Evaluation | Prototype | Place | Methods (and Tools) | Participants (Number) |
|---|---|---|---|---|
| Formative | Low fidelity | Museum | Wizard of Oz<br>Think aloud protocol<br>Observation (video and notes)<br>Interviews (semi-structured)<br>Questionnaire (UEQ) | Stakeholders (10)<br>Visitors (11) |
| Formative | High fidelity | Lab | Think aloud protocol<br>Observation (video and notes)<br>Interviews (semi-structured)<br>Questionnaire (UEQ) | Users (13) |
| Field study | High fidelity | Museum | Observation (notes)<br>Questionnaire (UEQ) | Visitors (56) |
| | | | Questionnaire (learning effect) | Visitors (15/56) |

### 5.1. Formative Evaluation (Museum)

The lo-fi prototype was assembled and installed in the museum's supporting room, where stakeholders and visitors were invited after their visiting experience. Before their interaction, they were divided at the two installation stations and provided with a small introduction. We also encouraged them to speak their thoughts while interacting with the system aloud (think aloud protocol). During the evaluation, the system interaction was performed using the 'Wizard of Oz' method. A human, in our case a team member, generated the system's responses to the user's actions without the user knowing that there was someone behind-the-scenes.

A total of 11 visitors and 10 stakeholders took part. Visitors' ages varied from high school students to middle-aged, whereas the stakeholders' average age was around 35 years old. We observed visitors' interactions while taking notes on the most persistent usability issues. Then, we proceeded to semi-structured interviews, mainly with the stakeholders and some enthusiastic visitors, to highlight their remarks. After that, we asked visitors to fill in the standardized User Experience Questionnaire (UEQ) [56].

In this evaluation phase, we wanted to examine the curators' and visitors' acceptance of the installation setup and test variations of the stations' interface: how it affects the user's viewing angles, the textual information, the possibility to avoid touch interaction with screens, and the placing area of the tangibles. Additionally, we needed feedback regarding the graphic aesthetics (Figure 12, left), such as colors and level of realism, especially from the museum's curators.

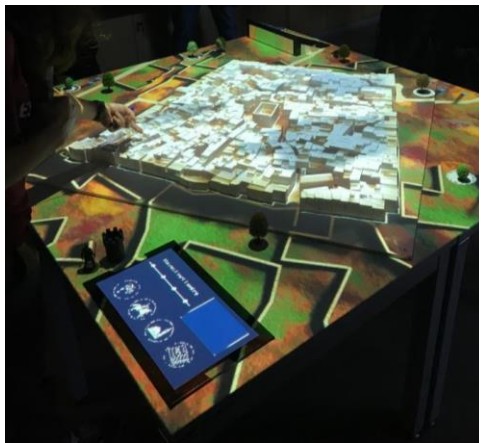
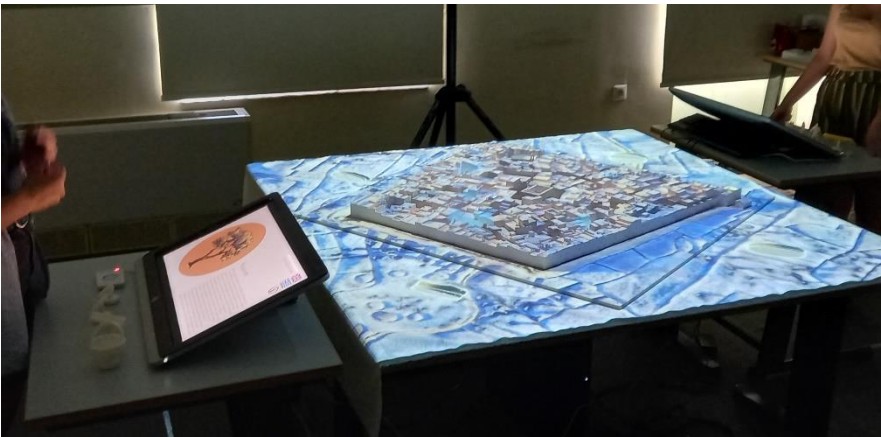

**Figure 12.** Screenshots from the formative evaluation at the Chios Mastic Museum's supporting room (**left**) and the laboratory evaluation (**right**).

The most significant outcomes of the formative evaluation can be summarized as four main points:

1. Most users preferred to use tangible objects to interact with the installation over digital touch keys;
2. Some users were confused by the appearance of the characters, which prevented their understanding of the story, and they would have liked a more realistic rendering;
3. Many users mentioned the possibility of augmenting the experience with a narrative that would provide more information about the events;
4. A more meaningful and symbolic connection was discussed between the 3D-printed tangibles and their related content.

Finally, the results of the UEQ are presented in Figure 13.

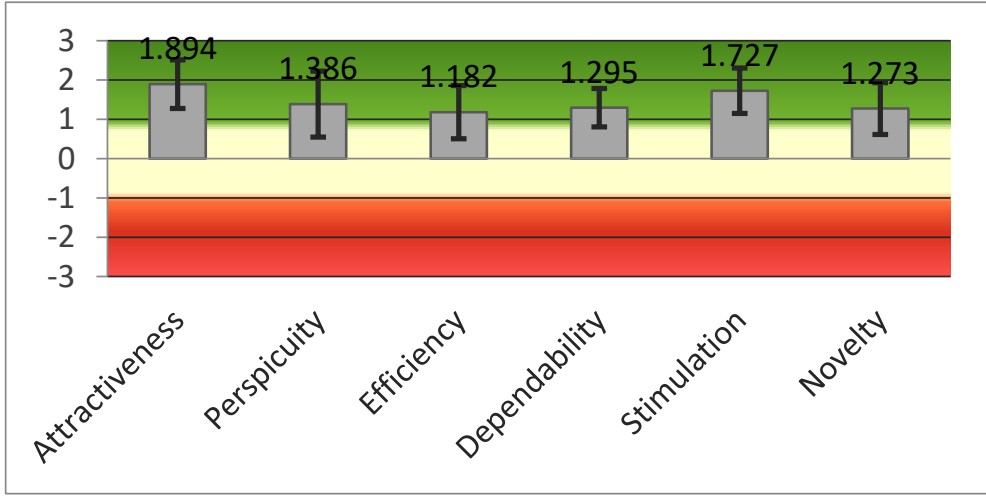

**Figure 13.** UEQ results of the formative evaluation.

### 5.2. Laboratory Evaluation

The lab evaluation took the form of a pilot study immediately before the final implementation of our interactive installation at the museum. In this phase, we focused on the experience the users gained through the improved and finalized version of the audiovisual narration. In addition, we examined the performance of our application and looked for any bugs or crucial issues that might hinder smooth interaction with the system. In addition, we wanted to evaluate the overall UX, even at a laboratory level, to determine how the

experience would be altered when inside the museum's context. Hence, lab users were also invited to fill in the UEQ.

In this evaluation phase, 13 users took part, and they were all undergraduate students of the Department of Product and System Design Engineering of the University of the Aegean with an average age of 23 years old. After a brief introduction, users interacted in pairs (and a triad) while being encouraged to think aloud. After that, they answered what they understood, what was tiring during the interaction, and what they would change, and then they answered the UEQ.

We found that the narration seems support the positive experience offered by the renewed displayed graphics (Figure 12, right). Furthermore, the characters, now combined with the sounds and the new animated illustrations compared to the lo-fi installation's version, did not confuse the users but enhanced the multi-sensory narrative, urging them to visit the museum. Finally, the UEQ provided some interesting results regarding the pragmatic and hedonic quality of the installation, even in a laboratory setting (Figure 14).

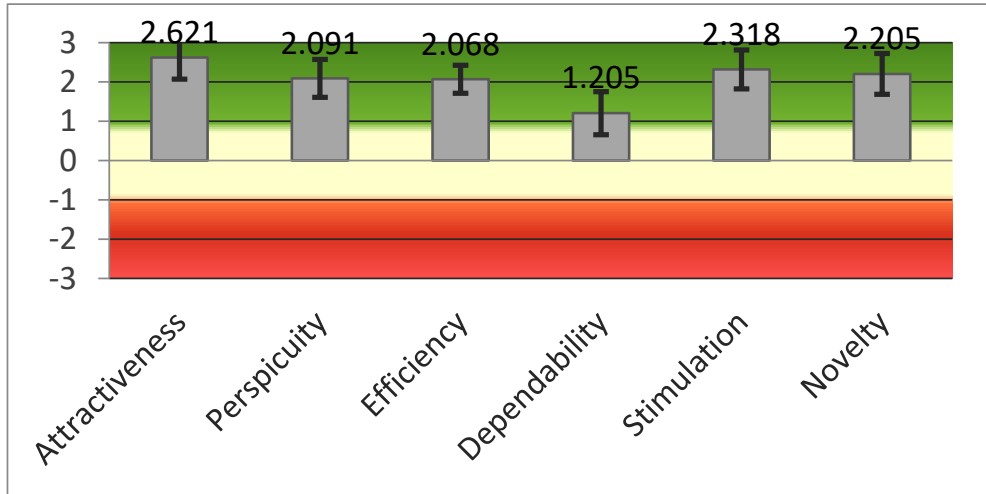

**Figure 14.** UEQ results of the laboratory evaluation.

### 5.3. Field Study

In the final study (field), we evaluated the UX of the installation in the museum setting. We installed the final version at a location at the end of the museum's visit, section E. left (Figure 1), where the visitors would have a concluding visiting experience, as advised by the museum's curators. We assessed the overall UX by providing users with the UEQ after interacting with the installation. In addition, we estimated the learning effect of the installation in a pre- and post-interaction questionnaire. The user participation was optional in the field study, and thus we considered participants to be users who provided data with either the UEQ, the learning effectiveness questionnaire, or both. In addition, we observed users while interacting with the installation and took notes. Finally, with some of the visitors, we had the opportunity to discuss their visiting experience and how the installation might affect it.

A total of 56 visitors interacted with the interactive projection mapping application at the Chios Mastic Museum over two months (Figure 15). Fifteen visitors from a total of 56 evaluated the learning effect of the interactive installation. To assess this aspect, we followed a different approach in recruiting users to answer the respective questionnaire since the testing required users to fill out the same questionnaire before (pre) and after (post) their interaction with the installation.

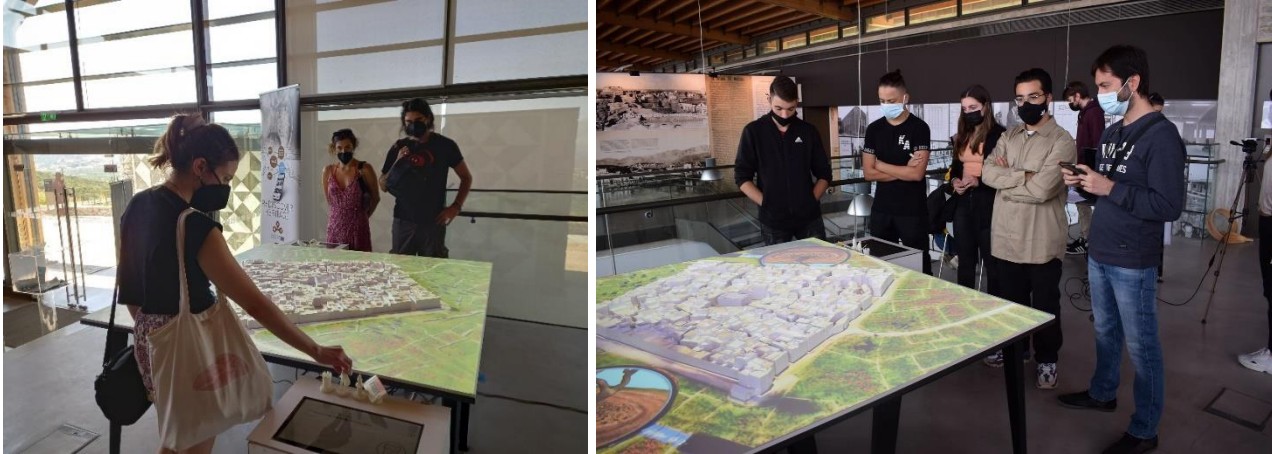

**Figure 15.** Screenshots from the field study evaluation at the Chios Mastic Museum: a triad of friend (**left**) and students with their teacher (**right**) interact with installation.

The learning effect questionnaire consisted of eight questions regarding the architectural evolution of the settlement with the historical events, the cultivation tools, and activities, and their particular names. Considering that the interactive installation is placed at the end of the museum visit, we assumed that visitors would answer the pre-interaction questionnaire by recalling what they had already seen and read during the visit to the related exhibits. Therefore, the answers to the post-interaction questionnaire and their scoring variables helped us to understand which aspects of the museum experience regarding the sections of mastic 'cultivation' and 'management' were enhanced in terms of deeper understanding and knowledge about the exhibits. The UEQ and the learning effectiveness questionnaire results are presented in Figure 16. and Table 3, respectively.

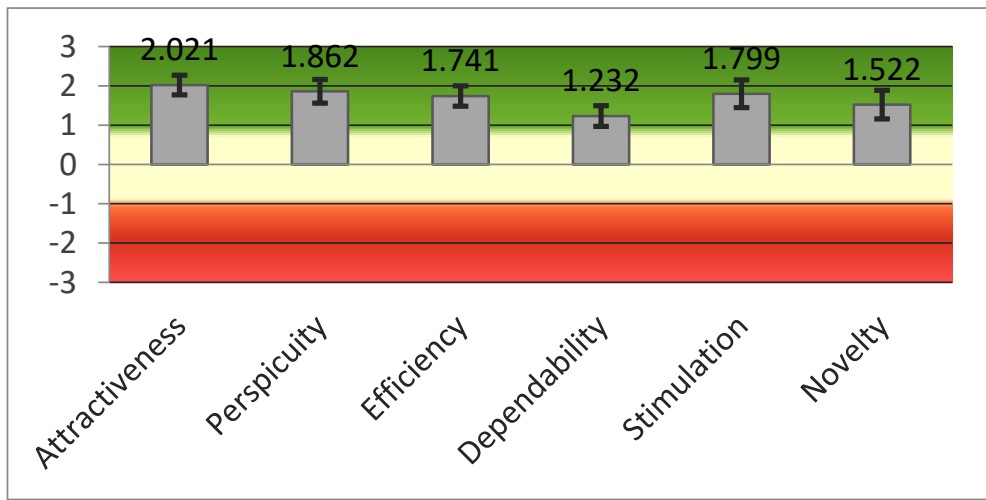

**Figure 16.** UEQ results of the field study.

**Table 3.** Results of the pre- and post-interaction questionnaire regarding the learning effect of the installation and *t*-test result.

| Pre-Test | Post-Test |
|---|---|
| 14.87/20 | 16.47/20 |
| *t*-test: *p*-value = 0.073 (>0.05) | |

In the field study, we observed visitors' acceptance of the overall appearance and interaction with the projection mapping installation, as manifested by the UEQ results. Nevertheless, many visitors appeared reluctant at their initial actions, i.e., grasping a tangible and placing it at the respective area of the station. Another critical issue observed was that some users were confused with the touch screens displaying the textual information. This seemed to urge them to either use touch input or place the tangible asset onto the respective screen's icon instead of the activation area.

All visitors were excited by the vivid illustrations, the audio narration, and the surrounding sound the installation presents. They were eager to pass through all the eight stories, and once they realized the stories' alternation passing from one station to the other, enhanced communication among them started to emerge. They began exchanging signals of who would place the next tangible asset while being set at the opposite sites of the installation. Another interesting outcome, and our initial requirement, related to the spatial affordances of projection mapping, is that many visitors walked around the installation to look more carefully at the projections inside the model's building blocks.

From discussions with more enthusiastic and less hasty visitors, we obtained an image of how aspects of the installation and its content affected their overall visiting experience. All visitors agreed on a recapitulating and educating experience with the installation since they could recall many aspects that they had already come across during their visit. However, in the *t*-test applied to the respective learning effectiveness questionnaire results, the *p*-value of 0.073 (>0.05) does not represent statistical significance. This could relate to the small user sample obtained for the particular testing or that some of the answers the users had to select were not put in a definite manner, or they might have been similar. However, some visitors realized the highlighting of the turret belonging to the fortified coastline system, which was only referenced in the video regarding the settlements' architectural evolution. Other visitors also asked us questions about the settlement and where it is located, showing eagerness to visit it. This is another interesting outcome of our installation, closely related to the overall museum's scope in raising awareness and recognition of a principal tangible cultural aspect, the mastic villages, which is the main focus of the Management section (D).

## 6. Discussion and Conclusions

Our interactive installation's design approach and evaluation studies reveal several design issues and considerations in the context of projection mapping applications in museums that aim to showcase ICH. The most apparent findings seem to confirm the advantages of SAR applications in museums in terms of their attractiveness and stimulation. In relation to the iterative evaluation process presented in this paper, we gain a clear view of the attractiveness of our installation, even in a laboratory setting or even at the lo-fi version of the installation during the first evaluation phase. This finding confirms the related research arguments that SAR holds the element of surprise by being attractive and appealing while keeping the users' stimulation levels at a high average [38,39,46]. From an observational point of view, we also report on accompanying enthusiastic users' reactions from all ages and backgrounds. Although our installation was placed in a bright environment, this did not hinder the element of amazement, and users showed eagerness to engage and follow the projected narration.

As far as user interface is concerned, there are a number of design issues related to the tangibles, the stations' interface, and the guidance and navigation provided to users. Tactile interaction through tangible museum replicas [29] was one of our primary decisions. However, we also wanted to test whether users might be keener to engage in touch interaction [57] during the first evaluation. Although the concept of having symbolic 3D-printed replicas that beautifully blend with the 3D-printed scale model in an aesthetic tone sounded valid from the formative evaluation, it seems that when more interaction affordances are offered to the user, it is possible to confuse input signals. Indeed there is research [38–40] indicating that when more information or guidance is added to the supporting SAR inter-

face, there is a chance of confusing, misleading, or even overwhelming users. We noticed incidents of frustration while visitors tried to figure out how the projections were triggered and how the touch screen responded, which is not encouraging, especially for older users who might stop trying after a while. This raises a concern about the various input affordances provided to the users and how re-designing users' navigation could overcome this problem. As seen from our evaluations and related previous research [53], the concept of keeping an iterative assessment of design issues improves the UX. Thus, it could be argued that our installation might not have reached its potential yet. Therefore, interaction affordances should be visible and clear to guide users to desirable actions, especially the initial ones, without the help of any dedicated museum personnel or an exhaustive guiding list. This is an issue for further research.

The collaborative and communicative advantage of projection mapping seems to affect our installation positively. Visitors usually interacted in pairs or groups, such as families, friends, or organized groups, and had the chance to engage with the projections in a synergetic turn-taking interaction, as suggested by our installation. The spatial arrangement also urged users to look around the model and inside the narrow folds that the settlement presents. This was a supplementary outcome to promote the architectural heritage of the place concerned. It also constitutes another appealing feature of SAR applications and one of our main initial requirements regarding the spatial affordances such installations can provide to users. Given that this type of technology offers a joint interactive opportunity among visitors, it would be meaningful to be used also in the museums' social context to facilitate a collective understanding of architectural heritage-related transformations [35].

The installation's numerous spatial configurations and projection capabilities made our design space vague. In a design space where every small decision affects the operation of the whole system, avoiding coding at the early stages helped us to reach the final result faster and more efficiently. This became apparent from our formative evaluation, where we used the 'Wizard of Oz' prototyping technique. It was important to take the first critical decisions regarding users' acceptance of the installation and its concept, the illustrations' appearance and aesthetics regarding the museum's curation. From this point of view, we confirm related research [58] that hints at the importance of semi-functional prototypes to repetitively improve interaction metaphors and techniques. When the prototypes reach their final version, then designers can test the system's feedback and users' understanding. In addition, the introduction of the 3D model into a simulation video mapping software reduced the time-consuming laboratory spatial testing configurations and the various projector's calibrations for the whole setup. In the meantime, the projection mapper and the graphic designer had the opportunity to collaborate on a digital level, experiment, and reach conclusions on the projector's technical aspects, such as the optimal viewing angle, distortion, and distance, and the advanced 2D and 3D graphic illustrations that smoothly fitted onto the 3D-printed model later.

Our 3D model's plain and white version was intended to simulate the original exhibited model of the settlement by removing its geographical constraints, such as the hills and other engraved architectural details, and giving thus the opportunity to focus on the mastic village in general and with a minimal approach. Moreover, it remains representative, meaning that it does not focus on the Olympoi settlement per se as the exhibited model but tries to abstractly incorporate all the mastic villages' histories and architectural evolutions, which are virtually the same for all of them. In this way, we promote all the settlements' histories and cultural heritages and shift the main interest to the ICH of a place. In addition, to follow up on the above remark on the users' eagerness to visit the settlement, the questions they asked were general and related to the location of the settlements and whether they still kept these architectural features. This could indicate an effective promotion of the ICH and its connection to the real place. The museum, in our case, stands as a heritage curator by collecting and representing a unique ICH. At the same time, it dedicates a specific section (Management, D.) to the tangible, architectural aspects of the heritage exhibited. It is noted that the "interpretation of the historical management of such a unique product and how it

affected the rural and inhabited landscape of the southern region of Chios and its settlements" constitutes the museum's second goal. Therefore, and given the relevant qualitative data, this goal seems to be boosted by our interactive installation, which appears to stand as a heritage mediator [41] between the exhibited ICH and the place concerned. The intended visitors' visits to the respective places, the mastic villages, that manifest aspects of the ICH showcased inside the museum could be another element of sustaining cultural connections and bindings between the museum and the area's communities, which is desired according to the newest museological research [13,14,17]. Furthermore, interactive documentaries, like the indicating in the present paper one, that applies non-linear storytelling techniques to freely engage the visitor with the heritage history, can be used in other media forms to ensure its digital sustainability [59,60].

To conclude our findings, the related applications in [35–37] are primarily focused on architectural features of the cultural buildings and try to reveal the essence and value of the place mainly via exploration. Our application tried to incorporate a holistic approach to connect architectural heritage with ICH via narration and deliver to visitors an interactive reflecting documentary in a representative space inside the museum, thus integrating all the three museum's goals. The introduced in this paper installation interposes the other two museum's aims, the first and the third, as it links the history and architecture with the traditional skill from its initial conceptualization. It also illustrates the promotion of mastic's cooperative exploitation by the "breaking" of the enforcing historic walls, imposed by the island's rulers, and the spread of the mastic gum sheet all over the village. In this kind of projection, the representative architectural model of the place integrates its ICH. It allows the latter to be alternatively illustrated, spatially perceived, and spiritually layered upon its object of reference. This application paradigm could work in similar curatorial contexts where ICH is decisively related to its cited place.

Nevertheless, our field study has some limitations concerning the participants' sample and the systematic recruitment of users to provide the appropriate data. The evaluation was deployed for a limited time in a real-life busy environment, where not all museum visitors have the willingness to participate or be interviewed. In addition, the learning effectiveness questionnaire cannot provide information related to the assessment of the communicated intangible values. However, many visitors seemed to have their awareness increased and to reflect on the place and its location and wonder whether the model refers to an old or new version of the represented settlement. These could indicate that some of the values between architectural and intangible heritage are communicated. However, we cannot strongly assume this, since it is mostly based on our qualitative data (observation, visitors' remarks). We believe an orderly triangulation to obtain results towards this direction is needed with more focused and, ideally, standardized questionnaires to provide quantitative data on the approach to communicating values.

Overall, this paper has explored a conceptual design approach to connect the architectural and intangible cultural heritage of a place with engaging and vivid audiovisuals projected on a scale model of a historic settlement. We have presented the design, development, implementation, and iterative user evaluation of a projection mapping installation that displays historical events and seasonal cultivation activities as part of the ICH of a place on a 3D-printed scale model of a representative historic settlement of Southern Chios exhibited inside a museum. The installation affords a tangible interaction to activate video projections that are presented in a storytelling manner, with a double set of 3D-printed artefacts placed on the opposite sides of the scale model to trigger either the historical events on the island or the cultivation activities the people perform throughout the year. Considering that the installation is positioned at the end of the museum's visit, we assumed that visitors would have a concluding opportunity to recapitulate and reflect on the intangible values the museum is aiming for the visitor to perceive. Therefore, our iterative design approach has been composed and processed in the context of this hypothesis. The final field study found that visitors raised their awareness and recognition of the tangible cultural aspects conveyed, urging them to visit the settlements. Moreover, the iterative user evaluations

at the lab and the museum identified several design and implementation issues for other designers, curators, and heritage professionals to consider for similar projection mapping applications in related museum contexts, where previous research had not extensively shed light before. Future research is needed, however, to further understand how such installations perform in terms of impacting the visiting experience in the long term and how they differentiate from similar low-tech approaches.

**Author Contributions:** Conceptualization, V.N.; Methodology, V.N., P.P. and S.V.; Software, P.P., V.M., D.K. and S.V.; Supervision, V.N.; Writing—original draft, V.N, and P.P.; Writing—review and editing, S.V., P.C. and P.K. All authors have read and agreed to the published version of the manuscript.

**Funding:** This research has been co-financed by the European Union and Greek national funds through the Operational Program Competitiveness, Entrepreneurship and Innovation, under the call RESEARCH–CREATE–INNOVATE (project code: T1EDK-15171).

**Institutional Review Board Statement:** Not applicable.

**Informed Consent Statement:** Informed consent was obtained from all subjects involved in the studies.

**Data Availability Statement:** The data of this research can be accessed under the ODC-BY 1.0 license after contacting the corresponding author. According to the ODC-BY 1.0 you are free to share, create or adapt the data if you attribute its use by citing this manuscript.

**Acknowledgments:** We thank the former Mayor of Chios and architect Manolis Vournous for his valuable information regarding the settlements and his guided tour of the Olympoi mastic village, and the Chios Mastic Museum's book guide by Anna Kallinikidou for the vital historical analysis. Finally, the corresponding author wants to personally thank Christina Mertzanidou for proofreading this paper.

**Conflicts of Interest:** The authors declare no conflict of interest.

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
