# Peer review of "Conveying Intangible Cultural Heritage in Museums with Interactive Storytelling and Projection Mapping: The Case of the Mastic Villages"

_heritage, doi:10.3390/heritage5020056_

Round 1

Reviewer 1 Report

I have to congratulate the authors for this deep work. The paper is well referenced, it is clear, completed, and keep the reader interest. Just some comments, as try to synthesize the table 5 in another kind of graphic because there are only 4 data to show. The different steps of the metodology can be explained from the beginning: the different phases, the size of the sample, the age, the time of use. A data I missed is the time that is required to use this installation. For the rest, a very good job.

Reviewer 2 Report

Introduction

Intangible Cultural Heritage is well explained in the opening section. The authors make a compelling argument in favour of moving away from normative, tangible heritage, towards the anthropocentric in a way that rightly emphasises human activity and culture as it has evolved over time. The depictions of how digital technology has empowered ICH and how museums are developing more interactive and engaging exhibits are both clearly presented and accurate. The potential of Spatial Augmented Reality in this domain is put forward very well. The review of existing use cases is very detailed could potentially be trimmed down a little, but it does offer a fair representation of how such SAR systems have been of benefit. It is quite a while into the article before the limitations and challenges of SAR are discussed and an earlier presentation of a problem statement could be helpful.

Design and development

This section is consistently thorough in its explanation of the system and how it features within the museum exhibit use case. The hardware configuration, interface design and user-interaction are all very well described. The authors rationale for ‘creatively exaggerating’ certain aspects of the installation to ensure the required narrative is effectively communicated, I found to be convincing.

Evaluation

The use of iterative design with formative testing even at the lo-fi prototype stage, is a solid approach to developing MR. The three-stage approach to testing, from lo-fi field, to lab, to final field, is most certainly a solid level of evaluative rigour. What I understood to be the think-aloud protocol is an appropriate method in this context, particularly as there are multiple simultaneous users. I hadn’t actually heard of the Wizard of Oz method before, but it is well-described and appears appropriate as an efficient response to the testing requirements. Whilst I don’t take any issue with the use of subjective user-reporting for MR evaluations, ultimately, I always hope to eventually see more objective (mainly, performance-related) measures being utilised so I was very pleased to see the inclusion of the ‘learning effect questionnaire’. Using a pre/post-test is appropriate for demonstrating effectiveness though there is the clear limitation of no control. Whilst the results may provide us with some evidence of the system’s effectiveness, they cannot tell us if that effect is meaningfully greater than a much more low-tech (and low-cost) equivalent solution.

I feel that the discussion is balanced and is a fair representation of the results, with no notable exaggeration and full disclosure of issues and challenges.

Line 611: is it “few users” or “a few users”?

Overall

Whilst the system as described is not a revolution in MR museum installations (and it doesn’t pretend to be), bringing together tangibles, NFC, projection displays and digital displays within a multi-user experience is an interesting blend of techniques and technologies. The findings of the evaluation do feel rather familiar and reflect other studies in many places, but I would argue the commitment to a thorough and multi-stage evaluation does distinguish this work from many of its peers. Whilst I did have some issues with the evaluation method (see above), I would describe the academic soundness as very good and the overall communication, inclusive of structure, clarity, illustration and coherence, was excellent. I would very much like to see future studies of this type evaluating MR systems directly against traditional installations, ideally using more mid-to-long term user-impact metrics, to really discover the added value of this technology for this particular application. That said, I feel this paper is a fair contribution to scholarship and would recommend it for publication.

Reviewer 3 Report

Dear author/s
Thank you for giving me the opportunity to read your manuscript. I read your manuscript with great interest. Your manuscript is well structured and organized and partially follows the instructions of the journal. However, the manuscript needs minor or even major changes that will significantly improve it. Below you will find some minor or major points in the manuscript which need clarification, refinement, reanalysis, rewrites and/or additional information and suggestions for what could be done to improve it.

Section 1 (Introduction) needs a little revision. Some information and/or points are missing or unclear, and should be better included or rewritten (e.g., objectives of the study and/or hypotheses or research questions, etc.-which would be good to be numbered). To help you, here is a list of items that can be included in this section:
-What is the importance of making this research/contribution that it brings to the literature in the field?
-Why should readers be interested?
-What problem/ gap resolve/fill this research?
-To fill this gap (resolve this problem) what solution/intervention/benefits does this research bring? (In other words, how the proposed study will remedy this deficiency/gap/problem and provide a unique contribution to the literature)
-What is the research question which addresses the purpose of the research?
In summary, although some of the above are mentioned, unfortunately they are not clear. Additionally, if you consider it necessary, you can further expand this section with more literature data to support it. Please make the necessary adjustments.

Section 5 (Evaluation process and results) is also missing some points and information (e.g., the type of methodology, information about the methods you used, reliability, the consent protocol, etc.) or is unclear. Although some of them are mentioned, this section needs minor revision.

Section 6 (Discussion) seems to be well argued. Please extend this section a little more. In particular, please cite more of the journal papers published by MDPI where possible.

Section 7 (Conclusions) is very simple and looks poor. Please extend this section or you could merge it with Section 6 (Discussion) and rename it to "Discussion and Conclusions".

In conclusion, certain sections of the manuscript begin or end abruptly, which may reduce the reader's attention or interest. I would suggest the author/s could consider including some introductory paragraphs regarding the content of each section, in order to give the reader an idea of what to expect. Additionally, some discrepancies were observed between the present manuscript and the required template. The author/s also has/have a list of useful references, it may also make the manuscript more interesting if the author/s could use references below:
https://doi.org/10.3390/su13031193
https://doi.org/10.1016/j.culher.2018.07.016

As a final comment, kindly read your manuscript again with a clear mind and make the necessary corrections. You may need to move some parts to other sections for there to be a logical flow. Moreover, kindly check for grammatical errors, and new publications that could form part of the manuscript.

Round 2

Reviewer 3 Report

Dear author/s,
I have read again with much interest your revised manuscript.
The manuscript has been significantly improved.
Therefore, I consider the current version to be suitable for publication.
Congratulations on the effort made to improve the work!